# Factors Associated with Advance Directives Documentation: A Nationwide Cross-Sectional Survey of Older Adults in Korea

**DOI:** 10.3390/ijerph19073771

**Published:** 2022-03-22

**Authors:** Boram Kim, Jiyeon Choi, Ilhak Lee

**Affiliations:** 1Division of Health Policy, Bureau of Health Policy, Ministry of Health and Welfare, Sejong-si 30113, Korea; boramkim822@gmail.com; 2Division of Medical Law & Ethics, Department of Medical Humanities and Social Sciences, Yonsei University College of Medicine, Seoul 03722, Korea; jeons19@yuhs.ac; 3The Asian Institute of Bioethics and Health Law, College of Medicine, Yonsei University, Seoul 03722, Korea

**Keywords:** advance directives, advance care planning, life-sustaining treatment, end-of-life care, older adults, Korean

## Abstract

Advance directives (ADs) can support autonomy in making healthcare decisions and minimize unnecessary discomfort during the treatment process at the end of life (EOL). This study aimed to investigate the factors that influence AD documentation among community-dwelling older adults. We used data from the National Survey of Older Koreans which was conducted nationwide in 2020. Data from participants aged 65 years or older were extracted using stratified multistage cluster sampling and the survey was conducted through one-on-one interviews. A total of 9920 older adults were included in this study: 421 respondents (4.7%) claimed that they had prepared for AD. Multivariable logistic regression analysis showed that being 75 years or older, having higher educational attainment, higher income, having any chronic disease(s), being screened for dementia in the past, being against futile life-sustaining treatment, taking a lecture on death and being registered for organ donation were positively associated with AD. Furthermore, when health was rated as average, it was associated with reduced odds of AD documentation compared when health was rated as good. These results suggest that more targeted efforts are required to promote EOL discussions among older adults.

## 1. Introduction

Advance directives (ADs) are legal documents according to the Act on Hospice and Palliative Care and Decisions on Life-Sustaining Treatment for Patients at the End of Life (LST Decision Act) in South Korea. This allows competent patients to designate the kind of medical procedures they do not want to be treated with if they subsequently become unable to decide. ADs are expected to be effective in respecting the autonomous decisions of the elderly and the seriously ill, especially in cases where patients do not want to undergo futile treatment at the end of their lives [1,2]. It is critical for older adults to retain control of the level of care they receive through the execution of ADs [3].

In South Korea, the LST Decision Act has entitled competent adults (over the age of 19) to write ADs at registry agencies designated by the Ministry of Health and Welfare under the system for decisions to forgo LST since 4 February 2018. Approximately 90% of all AD writers in Korea are adults aged 60 years or older [4]. After writing the ADs, they show a positive shift in thinking when they understand they should cease worrying about death and rather prepare for it [5]. These results are similar to those reported in previous studies showing that many elderly people can decrease the burden on their families, and that ADs facilitate family members’ choices [6,7]. If older adults write an AD, they can maintain their autonomy and prepare for their end of life (EOL) [8].

To provide useful support for the elderly who want to make treatment decisions in advance, it is necessary to overcome the lack of robust data on the characteristics and motives for writing AD. This study aimed to investigate the factors associated with AD documentation among community-dwelling older adults. We hypothesized that demographic factors, health status, personal experiences, and social relationships would be related to EOL preparedness and AD documentation, providing insights into facilitators or barriers to participation in advance care planning (ACP) among older adults. By evaluating the factors associated with the presence of AD, it may be possible to provide suggestions to promote EOL discussions and AD preparation among the general public.

## 2. Materials and Methods

### 2.1. Data Collection

This study used data from the 2020 National Survey of Older Koreans. The survey was conducted nationwide by the Korea Institute for Health and Social Affairs and has been conducted every three years since 2008 to gather the basic information necessary for policymaking on senior-related issues [9]. The target participants of the survey were community-dwelling adults aged 65 years or older. Seniors who were hospitalized or living in nursing homes were not included in the survey because of difficulties in ensuring the representativeness of the target population. Participants were selected using stratified multistage cluster sampling based on the 2018 Population and Housing Census data. Sampling areas were stratified first by 17 districts, then again by urban–rural classification, and finally by house type [9,10]. A total of 903 sampling areas were extracted using this sample design. For the sampling distribution from sampling areas, the proportional distribution method was applied based on the size of the senior population in the census data. Various weights were applied to the survey data to ensure estimation accuracy [11]. A preliminary survey was conducted with 67 people from June to July 2020, and the implications were applied to the main survey model. The main survey was conducted between September and November 2020 [12]. Data were collected through one-on-one interviews using a tablet-assisted personal interview (TAPI) with the assistance of 169 interviewers. The interviewers visited the households in the assigned survey areas, and were assisted in this through the cooperation of the community service center staff. Visits were attempted up to three times if the participant was initially not at home. Although there were a few proxy responses in the survey, those data were not included in this study. Before the survey was conducted, the participants were informed that they would be given a small gift after completing the survey. Project supervisors monitored the process by randomly calling more than 40% of the participants and verifying their responses after the week’s survey results were collected [12]. They tried to call the participants up to three times if they did not initially respond to the phone call. Participants were asked whether they actually participated in the survey, and whether the survey was conducted according to the correct procedure.

This study was approved by the Institutional Review Board of Yonsei University Health System (IRB approval number: 4-2021-1078; date of approval: 28 September 2021).

### 2.2. Measurement

The sociodemographic characteristics examined in this study were gender, age, marital status, education level and annual income. We investigated the factors related to health status such as the number of chronic diseases, hospitalization history, dementia screening history and self-rated health. Participants were asked about chronic diseases that they had been suffering from for over three months after receiving a diagnosis from a doctor. They selected those applicable to them out of the 32 disease classifications and the numbers were then counted. The participants were also asked whether they had been hospitalized for an illness or injury in the past year. We investigated whether older adults had been screened for dementia in the past two years. In Korea, anyone older than 66 years can receive a dementia screening test as part of their National Medical Check-Ups, every two years for free. In addition, people over the age of 60 can receive free dementia screening at 256 dementia support centers nationwide which are operated by the government. There are shared characteristics between the preparation of AD and getting a screening test for dementia regarding preparation for deterioration in the decision-making ability of older adults. Self-rated health was measured on a five-point scale from “very good” to “very bad”. The responses were classified into three categories: good, average and poor. Participants were asked whether they had cared for a sick family member, including accompanying them to the hospital within the past year. Respondents were asked if they had participated in religious, social, leisure, and cultural activities during the past year to assess social relationships as a motivating factor or source of information. We asked whether the participants followed a specific religion, and if so, how many times a week they participated in these religious activities (worship, volunteer work and social gatherings). Social group activities were defined as participation in clubs, alumni meetings, and political or civic organizations. Participants were also asked if they had any experience using facilities such as senior citizen centers, welfare centers, and public cultural facilities, which are common places for leisure and cultural activities for older Koreans. To investigate the accessibility of information through digital devices, we asked whether the participants used at least one smartphone, tablet, laptop, or computer. Attitude towards LST was assessed by asking participants, “What do you think about providing LST even if you are unconscious or find it difficult to survive?” Responses of “strongly agree” to “strongly disagree” were adopted (5-point scale), and responses of “strongly disagree” and “disagree” were classified as having an attitude that did not agree with the futile LST. Participants responded to their preparations for their EOL and death: experience of attending a lecture on death and organ donation registration, respectively. As an outcome variable, participants were asked whether they had AD.

### 2.3. Statistical Analysis

We analyzed the procedures using weighted factors based on a complex sample design of the survey. The participants’ baseline characteristics are expressed as numbers (not weighted) and percentages with standard errors (weighted). The associations between various factors and the presence of AD were examined using univariate logistic regression. Potential variables associated with AD completion in the univariate analysis (*p* value < 0.10) were included in the multivariable analysis. Multivariable logistic regression was used to investigate the factors related to AD. Adjusted odds ratios (AORs) and 95% confidence intervals (CI) were calculated for multivariable-adjusted models, and a *p* value < 0.05 was considered statistically significant. All analyses were performed using SPSS version 28.0 (SPSS Inc., Chicago, IL, USA).

## 3. Results

### 3.1. Characteristics of the Participants

Of the 10,097 survey participants, 9920 were included in this study after excluding those who responded by proxy. The participant characteristics are presented in Table 1. There were more female (56.9%) participants. The largest age group was 65–69 years (33.6%) and the number decreased as age increased. More than two-thirds of participants were married (67.5%), more than half of the participants had received a middle school education or above (58.2%), and 54.6% had an annual income of less than KRW 10 million. In terms of health status, more than half of the participants (54.5%) had two or more chronic diseases. Hypertension was the most prevalent chronic disease among the older adults (56.8%), followed by diabetes (24.2%), dyslipidemia (17.1%), osteoarthritis or rheumatoid arthritis (16.5%), and lower back pain or sciatica (10.0%). More than half of the participants (57.4%) had not been screened for dementia within the past two years. Most participants had not been hospitalized in the past year (92.9%), and answered that their health status was good or average (80.2%). Almost half (49.0%) had taken care of a sick family member in the past year. More than half of the older adults had participated in religious activities (53.4%) but the inverse was true for those that had participated in social group activities (45.2%) or leisure/cultural activities (38.4%). Digital devices such as smartphones were used by more than half of the participants (53.7%). Most participants responded that they did not agree with futile LST (85.7%). Participants who had prepared for their EOL and death were investigated as follows: experience of attending a lecture on death (2.7%) and organ donation registration (3.4%). Among the participants, 4.7% had previously completed an AD.

### 3.2. Factors Associated with the Presence of AD

Table 2 shows the results of the univariate analysis of the factors associated with the preparation of ADs. All variables except social group activities and the use of digital devices were associated with the presence of ADs (*p* < 0.10).

All variables found to be associated with the presence of ADs by univariate analysis were included in multivariable logistic regression analysis. Controlling for all other variables in the model, AD was associated with age, education level, income, number of chronic diseases, dementia screening test, self-rated health, attitude towards futile LST, experience of attending a lecture on death, and organ donation registration (Table 3). Considering the sociodemographic aspects, participants who were aged 75 years or older, had higher educational levels, and those that had a higher income were more likely to have ADs. Participants with chronic diseases were more likely to have ADs than those without. Dementia screening history and attitude towards disagreeing to futile LST were statistically significantly associated with the presence of ADs. Participants reported that those with an average health status were less likely to have ADs than those with a good health status. Those who attended a lecture on death were nearly five times more likely to have ADs than those who did not attend. Those who registered their decision to donate organs were eight times more likely to have AD than those who had not.

## 4. Discussion

Our study describes the association between ADs and related factors among community-dwelling older adults. Approximately 5% of the participants reported that they had ADs. In Korea, all ADs are legally registered by the registry agencies designated by the Ministry of Health and Welfare. By 2020, there were approximately 700,000 registered ADs for Koreans aged 60 or older [4]. This accounted for approximately 5.6% of the total number of individuals in the same age range, which is comparable to the results of our study. The prevalence of AD found in our study was relatively lower than that in the United States (46–51%) [13] or Australia (14%) [14], which is noteworthy because it resulted from two years of legislation on AD documentation in Korea.

### 4.1. Socio-Cultural Characteristics of Older Adults Who Write ADs

The results of our study were consistent with those of previous studies that revealed that senior citizens aged 75 years or older [15,16,17], with a high education level [15,18,19] and a high income [17,20], were more likely to have ADs. As people reach an age when they often experience their own personal health crises or the death of others, they may become more motivated to prepare for the loss of decision-making capacity [21]. Several studies have revealed that education level and income status are closely related to health literacy and healthcare access, as well as awareness of ADs [22,23]. This can be interpreted as AD being written only when the person has a high understanding of it, although the concept has spread rapidly in Korean society.

Previous studies have reported that being female and widowed are predictors of ADs among older adults [14,15,24]; however, no such associations were found in our study. Since other studies have also reported no associations between ADs and gender or marital status [17,25,26,27], further research on sociodemographic factors is needed.

### 4.2. The Purpose of Writing ADs among Older Adults

Older adults with chronic diseases were more likely to have ADs than those without, which is consistent with the results of previous studies [3,14]. Older patients with comorbidities were more likely to have had prior discussions with their healthcare providers regarding their preference for EOL care [14,28]. Chronic diseases are characterized by a large burden of care because of their prolonged duration. Therefore, older adults with chronic diseases may have ADs to prepare for the burden on their families due to a disease.

Participants were also more likely to have ADs if they had been previously screened for dementia. Previous research findings suggest that older adults who have recently seen a healthcare provider for treatment or screening are more likely to have ADs [16,24]. According to these studies, people apparently do not think about ADs until they realize that they might need it. Contact with healthcare providers may be a factor that promotes the preparation of ADs, possibly reflecting the role of the primary care physician in facilitating ACP documentation [24]. Another explanation is that receiving a diagnosis of cognitive impairment triggers ACP as a means of preparing for future decisional incapacity [29].

### 4.3. The Motivation of Writing ADs among Older Adults

Our study showed that attitudes towards futile treatment was related to writing ADs, as participants who did not agree with providing futile LST were more likely to have ADs. This result is consistent with the results of previous studies showing that people who preferred palliative care rather than LST were more willing to make ADs in EOL care [30,31]. This suggests that developing ADs is closely related to people’s intention to control their own care and the right to refuse futile treatments [30]. Existing studies have reported that educational interventions are effective in assisting ACP implementation and AD completion among older adults [32,33]. Among the participants in our study, those who had received an educational program regarding EOL issues at various institutions, including welfare and nonprofit organizations, were also more likely to have ADs. Education for older adults guides them on how their wishes can be respected, thereby promoting autonomy and influencing their preferences for LST and attitudes towards ADs [34]. Educational programs for EOL and ADs must be publicly implemented. To support this, future studies on educational strategies to promote AD-related discussions are required.

Similar to previous research results [35,36], participants who registered for organ donation were more likely to have ADs. This suggests an opportunity to increase awareness and receptivity towards organ donation by including a brief segment on this issue during ACP discussions [36]. In some countries such as the United States and Australia, most advanced medical directive forms include organ donation sections, but not in Korea. In future research, it will be necessary to study how various topics and wishes for EOL should be discussed and recorded.

Univariate analysis showed that older adults with experience of caring for a sick family member were more likely to have ADs, but the multivariable analysis did not. Previous studies have reported that the experience of caring for people with a terminal illness or that require intensive care (e.g., mechanical ventilation) is associated with the presence of ADs [37,38,39,40]. These experiences were characterized by witnessing the painful death of a loved one, and people were motivated to participate in ACP as a means of limiting medical treatment to avoid unwanted interventions at EOL [37]. Respondents would choose to make advanced decisions to avoid being a burden for their future caregivers [30]. National health surveys in Korea did not include information on the disease severity of the families cared for by the participants; this needs to be included in future surveys.

### 4.4. Barriers against Writing AD in Older Adults

Barriers that limit the uptake of ACP have effects at the individual, interpersonal, provider and system levels [41]. Common barriers described at the individual and interpersonal levels were lack of awareness of ADs and relationship concerns. In this study, we investigated whether individuals’ knowledge and experience are related to having ADs. Although this study partially included family relationships at the interpersonal level, future research should consider that they may have various effects on EOL discussions and AD documentation. In Asia, including Korea, the family-driven decision-making culture acts as a barrier to the discussion of EOL. This is due to its taboo status among the elderly, making it difficult to lead to ADs documentation [31,42]. Regarding provider and system factors, it is necessary to consider medical and social environments. There have been few opportunities for medical personnel to experience ADs in the curriculum and medical practice, such as in Korea, where it was recently established in a legal form [43,44]. In addition, policy efforts to support the documentation of ADs have recently begun in Korea, and it is necessary to continuously investigate whether there are any barriers to accessing AD documentation at these levels.

### 4.5. Improvement for Barrier Resolution

Social relationships may broaden opportunities for sharing information about ADs and communicating perspectives and preferences about EOL care, which can lead to further discussions and documentation of ACP [22,24]. Although older adults mainly discuss EOL with family members [25,45], in reality, only a few do so because they do not want to burden their family members [46,47]. Furthermore, the number of older Asians living without children or living in single-person households has rapidly increased [48]. Social capital (e.g., alumni meetings, religious groups, sports clubs, cultural activity groups, political or civic organizations) is becoming an important health information resource [49]. Although our study did not show an association between social activities and AD documentation, which is inconsistent with other studies [50,51], future research should examine whether and specifically which social activities influence these individual decisions.

Internet use among older adults is rapidly growing, and our study showed that more than half of the participants used digital devices. Increasing older adults’ Internet use can improve access to health information and health self-management [52]. Prior studies have reported that people heard about ACP mostly through the media and less often through family or healthcare providers [31,53]. Contrary to the results of previous studies [19], an independent association between the presence of ADs and the use of digital devices in older adults was not observed in our study. Further studies are required to determine where older adults obtain information about ADs and how it leads to AD documentation.

### 4.6. The Implications and Limitations of the Study

To the best of our knowledge, this study was the first to examine the association between the presence of ADs and related factors based on a nationwide survey of older Koreans. Survey participants were recruited via a proportional stratified sampling method based on population census data. All the data were collected through one-on-one interviews with trained surveyors. Therefore, the results of this study can be better generalized to older adults than those of other studies, which were conducted using a sampling method with potential selection bias (e.g., convenience sampling, telephone surveys) or that obtained data from self-report questionnaires.

However, this study had several limitations. First, a cross-sectional design was used, and the results could not be interpreted as causal inferences. Information regarding when the participants completed their AD documentation in the past was not included. The time interval between the date the AD was prepared and the day the survey was conducted is unknown. However, since ADs were only legalized two years before the time of the survey, the interval was not expected to be long. Second, there may be important factors (e.g., experience of one’s own life-threatening illness) related to ADs that were not included in this study. Previous studies have reported that having suffered from a life-threatening illness is a predictor of the presence of ADs [28,39]. The experience of suffering from a life-threatening illness can influence decisions related to patient autonomy [39]. We included participants with a history of hospitalization in this study; however, no association with the presence of AD was observed. Prior studies have shown conflicting results regarding the association between a history of hospitalization and ACP documentation [16,17,24]. Because an individual’s hospitalization experience may vary depending on the severity of the illness, future research should investigate its association with AD. Finally, older adults in nursing homes or long-term hospitalizations were not included in this study because of difficulties in ensuring representativeness. Previous studies have reported that ACP interventions have beneficial effects, such as the avoidance of unwanted hospitalization and LST in nursing home populations [54], and future studies involving these people are needed.

## 5. Conclusions

Our study explored the factors associated with AD documentation in older adults. Some sociodemographic and health-related variables were found to be significantly associated with AD. Personal attitudes and experiences in preparing for EOL were identified as strong predictors of the presence of ADs. ACP aims not only to ensure autonomy in older adults but also to increase the use of palliative care, reduce stress during EOL, shorten hospital stays and improve communication between healthcare providers and healthcare proxies. ACP can also help families prepare for the death of a loved one, resolve family conflicts, and help with bereavement [55,56]. To promote the discussion of ADs among older adults, it is necessary to examine which past experiences they are influenced by, where they obtain information from, and what other behaviors they are concerned with when considering future healthcare decisions. Our study may have implications for future research because it provides information on proper approaches for older adults who need support for discussions related to EOL.

## Figures and Tables

**Table 1 ijerph-19-03771-t001:** Baseline characteristics of the participants (*n* = 9920).

Variables	Number ^1^	% (SE) ^2^
Sex		
Male	3971	43.1 (0.6)
Female	5949	56.9 (0.6)
Age (years)		
65–69	3511	33.6 (0.5)
70–74	2466	23.4 (0.5)
75–79	1956	22.8 (0.5)
≥80	1987	20.3 (0.5)
Marital status		
Currently married	5849	67.5 (0.5)
Widowed	3648	29.6 (0.5)
Divorced/separated/unmarried/others	423	2.9 (0.2)
Education level		
Illiterate or barely literate	1122	10.3 (0.4)
Elementary school	3309	31.6 (0.5)
Middle school	2330	23.4 (0.5)
High school	2654	28.8 (0.5)
College or more	505	6.0 (0.3)
Annual income (thousands of KRW)		
<5000	2440	25.2 (0.5)
5000–9999	3038	29.4 (0.5)
10,000–19,999	2066	20.2 (0.5)
≥20,000	2376	25.2 (0.5)
Number of chronic diseases		
0	1678	16.2 (0.4)
1	2922	29.4 (0.5)
2	2712	27.1 (0.5)
≥3	2608	27.4 (0.5)
Hospitalization for illness or injury in the past year		
No	9272	92.9 (0.3)
Yes	648	7.1 (0.3)
Screening for dementia in the past two years		
No	5716	57.4 (0.6)
Yes	4204	42.6 (0.6)
Self-rated health		
Good	4940	49.4 (0.6)
Average	3120	30.8 (0.5)
Poor	1860	19.9 (0.5)
Cared for a sick family member in the past year		
No	5477	51.0 (0.6)
Yes	4443	49.0 (0.6)
Religious activities in the past year (per week)		
No religion or 0	4719	46.4 (0.6)
<1	2204	21.7 (0.5)
≥1	2997	31.7 (0.6)
Social group activities in the past year ^3^		
No	5723	54.8 (0.6)
Yes	4197	45.2 (0.6)
Leisure and cultural activities in the past year ^4^		
No	6118	61.6 (0.6)
Yes	3802	38.4 (0.6)
Use of digital devices		
No	4826	46.3 (0.6)
Yes	5094	53.7 (0.6)
Attitude towards futile LST		
Agree to provide LST	1355	14.3 (0.4)
Disagree to provide LST	8565	85.7 (0.4)
Experience of attending a lecture on death		
No	9651	97.3 (0.2)
Yes	269	2.7 (0.2)
Organ donation registration		
No	9592	96.6 (0.2)
Yes	328	3.4 (0.2)
Advance directives documentation		
No	9499	95.3 (0.3)
Yes	421	4.7 (0.3)

^1^ Data were not weighted. ^2^ Data were weighted to yield nationally representative estimates. ^3^ Participation in clubs, alumni meetings, political or civic organizations, etc. ^4^ Any experience of using facilities such as senior citizen centers, welfare centers and public cultural facilities. SE, standard error; KRW, Korean won; LST, life-sustaining treatment.

**Table 2 ijerph-19-03771-t002:** Univariate analysis of factors associated with the presence of advance directives among older adults (*n* = 9920).

Variables	OR (95% CI)	*p* Value
Sex		
Male	1	
Female	0.70 (0.56–0.89)	0.003
Age (years)		
65–69	1	
70–74	1.13 (0.84–1.50)	0.420
75–79	1.40 (1.02–1.93)	0.036
≥80	1.17 (0.84–1.64)	0.358
Marital status		
Currently married	1	
Widowed	0.73 (0.56–0.94)	0.015
Divorced/separated/unmarried/others	1.33 (0.83–2.13)	0.242
Education level		
Illiterate or barely literate	1	
Elementary school	2.29 (1.45–3.60)	<0.001
Middle school	2.21 (1.37–3.56)	0.001
High school	2.88 (1.84–4.51)	<0.001
College or more	4.94 (2.86–8.53)	<0.001
Annual income (thousands of KRW)		
<5000	1	
5000–9999	1.21 (0.84–1.74)	0.310
10,000–19,999	1.73 (1.20–2.49)	0.003
≥20,000	2.04 (1.45–2.88)	<0.001
Number of chronic diseases		
0	1	
1	1.80 (1.22–2.65)	0.003
2	1.59 (1.07–2.35)	0.021
≥3	1.86 (1.27–2.74)	0.002
Hospitalization for illness or injury in the past year		
No	1	
Yes	1.83 (1.23–2.74)	0.003
Screening for dementia in the past two years		
No	1	
Yes	1.69 (1.34–2.14)	<0.001
Self-rated health		
Good	1	
Average	0.64 (0.48–0.84)	0.002
Poor	1.04 (0.77–1.41)	0.084
Cared for a sick family member in the past year		
No	1	
Yes	1.25 (0.99–1.58)	0.057
Religious activities in the past year (per week)		
No religion or 0	1	
<1	1.05 (0.78–1.42)	0.755
≥1	1.31 (1.00–1.70)	0.049
Social group activities in the past year ^1^		
No	1	
Yes	1.18 (0.93–1.48)	0.174
Leisure and cultural activities in the past year ^2^		
No	1	
Yes	1.22 (0.97–1.55)	0.094
Use of digital devices		
No	1	
Yes	1.03 (0.81–1.30)	0.823
Attitude towards futile LST		
Agree to provide LST	1	
Disagree to provide LST	1.48 (1.02–2.15)	0.039
Experience of attending a lecture on death		
No	1	
Yes	6.70 (4.77–9.42)	<0.001
Organ donation registration		
No	1	
Yes	11.14 (8.13–15.26)	<0.001

^1^ Participation in clubs, alumni meetings, political or civic organizations, etc. ^2^ Any experience of using facilities such as senior citizen centers, welfare centers and public cultural facilities. OR, odds ratio; CI, confidence interval; KRW, Korean won; LST, life-sustaining treatment.

**Table 3 ijerph-19-03771-t003:** Multivariable logistic regression analysis of factors associated with the presence of advance directives among older adults (*n* = 9920).

Variables	AOR (95% CI)	*p* Value
Age (years)		
65–69	1	
70–74	1.25 (0.90–1.73)	0.179
75–79	1.78 (1.21–2.60)	0.003
≥80	1.67 (1.08–2.61)	0.022
Education level		
Illiterate or barely literate	1	
Elementary school	2.24 (1.38–3.61)	0.001
Middle school	2.29 (1.32–3.99)	0.004
High school	2.88 (1.65–5.03)	<0.001
College or more	3.95 (2.08–7.48)	<0.001
Annual income (thousands of KRW)		
<5000	1	
5000–9999	1.24 (0.84–1.82)	0.283
10,000–19,999	1.52 (1.03–2.25)	0.036
≥20,000	1.70 (1.15–2.53)	0.008
Number of chronic diseases		
0	1	
1	1.77 (1.19–2.64)	0.005
2	1.55 (1.03–2.33)	0.034
≥3	1.96 (1.28–3.01)	0.002
Screening for dementia in the past two years		
No	1	
Yes	1.44 (1.12–1.85)	0.004
Self-rated health		
Good	1	
Average	0.64 (0.46–0.87)	0.005
Poor	0.99 (0.69–1.43)	0.084
Attitude towards futile LST		
Agree to provide LST	1	
Disagree to provide LST	1.55 (1.04–2.32)	0.034
Experience of attending a lecture on death		
No	1	
Yes	4.63 (3.13–6.84)	<0.001
Organ donation registration		
No	1	
Yes	8.43 (6.05–11.75)	<0.001

AOR, adjusted odds ratio; CI, confidence interval; KRW, Korean won; LST, life-sustaining treatment.

## Data Availability

Datasets from the National Survey of Older Koreans database are available for researchers who meet the accessibility criteria for obtaining confidential data. Researchers can apply for data on the Health and Welfare Data Portal operated by the Korea Institute for Health and Social Affairs (https://data.kihasa.re.kr, accessed on 25 November 2021).

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
