# Peer review of "Factors Associated with Advance Directives Documentation: A Nationwide Cross-Sectional Survey of Older Adults in Korea"

_ijerph, 2022, doi:10.3390/ijerph19073771_

Round 1
Reviewer 1 Report
Very good original article which covers the various factors which may affect decisions regarding the pursuit of advance directives documentation in the context of the korean population. I found the methodology easy to follow and the results well-presented. The discussion points were adequate and appropriate to support the findings from this study, and also addressed limitations of this study/ remaining barriers to AD decision-making amongst other issues which require further investigation in the topic.
There is no obvious issues I can pinpoint to when re-reading it critically again. There are only certain words used in the manuscript that in my mind I may have written differently (e.g. In abstract, Line 13 - use 'discomfort' instead of 'pain'), but I feel that the communication of the content is fairly clear and it was an enjoyable read which I think is of sufficient quality to publish in IJERPH.
Author Response
We appreciate your kind words. And we reflected your recommendation to choose the word discomfort instead of pain.
Reviewer 2 Report
This submission aimed to investigate the factors associated with Advance Directives documentation among community-dwelling older adults in South Korea as there is lack of robust data on the characteristics and motives of writing AD. This information has become particularly relevant to South Korea as the LST Decision Act has entitled competent adults to write AD since Feb. 4, 2018.
The study is well-formulated and clearly written, documented and analyzed. The Tables are useful, well-designed and easy to read. The limitations are thoughtfully considered. There are no major problems with this submission, it is timely and relevant. There are only minor issues, which will be mentioned line by line below.
Line by line suggested edits.
44-45 “If older 44 adults write AD, they can respect their autonomy to prepare for their end-of-life (EOL).”—need a reference for this (likely what is now reference 33).
76 Change “that data” to “those data”.
77-78 Please mention whether the participants knew in advance that they would be receiving a small gift to participate in completing the survey.
78-79 “Project supervisors monitored the process by calling more than 40% of the participants and checking their responses”—when did the project supervisors make these calls in relation to when the survey was conducted? How many times did the project supervisors try to call if the participant didn’t pick up the phone? What did the project supervisors check for when they made these calls?
148 Under “Education level”, the first entry likely should be “Illiterate or barely literate” as illiterate or literate includes all participants.
159 Under “Education level”, the first entry likely should be “Illiterate or barely literate” as illiterate or literate includes all participants.
180 Under “Education level”, the first entry likely should be “Illiterate or barely literate” as illiterate or literate includes all participants.
References
Please list abbreviated journal names for all journals as indicated in the Instructions for Authors.
As per the MDPI Reference List and Citations Style Guide, please use a semicolon after the page numbers and before the doi rather than a comma.
7 Please correct this reference. The publication, not the place of publication, should be italicized. Also, indicate that the publication is available online and the date that it was accessed.
8 Please include the doi.
9 Please include the doi.
10 Please correct this reference. The publication, not the place of publication, should be italicized. Also, indicate that the publication is available online and the date that it was accessed.
11 Please correct this reference. The publication, not the place of publication, should be italicized. Also, indicate that the publication is available online and the date that it was accessed.
20 Please include the doi.
30 Please include the doi.
47 Please include the doi.
48 Please include the doi.
Author Response
We do appreciate your thorough review and comments on the manuscript. We revised the manuscript reflecting your comments as attached.
- we cited on the 44-45 sentence. [8]
- the description of the data collection methodology was revised to state information given, and participants contact
- we corrected the education level category "illiterate and barely literate" as "illiterate and barely illiterate" to include all participants.
- we revised references to include Doi's and to comply citation rules.
Once again thank you for your helpful comments.
Reviewer 3 Report
Dear Authors, I have read your manuscript with interest.
The current manuscript titled: "Factors associated with advance directives documentation: A nationwide cross-sectional survey of older adults in Korea" represents an important analysis of evolving field of Public Health.
The title reflects the manuscript content and helps the reader navigate the article essence.
The abstract contains all the necessary information in a concise form.
The introduction section is clear and easy to read. It provides the basic overview of the current problem, well documented.
The author described in detail the methods used, patient group, biomarkers used, method of data extraction and the statistical methods used to process the presented data.
The result section is well written and detailed. For a better results understanding, the Author has attached tables.
In my opinion, these are the adjustments which should be made to increase the value of your manuscript:
- In title, change please all first words letters to capital.
- Make please the following expressions the same: Line 13 “end of life (EOL)” and Line 45 “end-of-life (EOL)”.
- Line 40: change please “Feb.” to “February”.
- Line 125: write “and” without italic.
- Line 133: after “65 to 69 years old”, add please “(33.6%)”.
- In Tables 1,2 and 3, change please “Literate” to “literate”.
- Please, indicate in more detail what chronic diseases the patients suffered from and present their percentage. This information is important.
- The manuscript contains some punctuation errors, please revise the text.
Author Response
We appreciate your insightful comment on the manuscript. We revised it to reflect your comments.
- We used capital letter for the title.
- We chose the expression "end-of-life (EOL)" in the manuscript
- We corrected typos, and added missing percentage.
- We added informations of the chronic diseases of the respondents.
- We re-edited the manuscript to correct it grammar errors.